# Evaluation of Microstructure and Abrasive Wear-Resistance of Medium Alloy SiMo Ductile Cast Iron

**DOI:** 10.3390/ma17051223

**Published:** 2024-03-06

**Authors:** Łukasz Dyrlaga, Renata Zapała, Krzysztof Morgiel, Andrzej Studnicki, Andrzej Szczęsny, Dariusz Kopyciński

**Affiliations:** 1Faculty of Foundry Engineering, AGH University, al. Adama Mickiewicza 30, 30-059 Krakow, Poland; zapala@agh.edu.pl (R.Z.); krzysmorgiel@student.agh.edu.pl (K.M.); ascn@agh.edu.pl (A.S.); djk@agh.edu.pl (D.K.); 2METALPOL Węgierska Górka, ul. Kolejowa 6, 34-350 Węgierska Górka, Poland; 3Department of Foundry Engineering, Silesian University of Technology, ul. Towarowa 7, 44-100 Gliwice, Poland; andrzej.studnicki@polsl.pl

**Keywords:** ductile cast iron, SiMo cast iron, silicon addition, molybdenum addition, abrasive wear resistance, HV hardness

## Abstract

Medium-alloy ductile iron with a SiMo ferritic matrix has very good heat resistance. The addition of chromium and aluminum also increases this resistance. This article presents the impact of chromium and aluminum on the structure of SiMo cast iron, especially their impact on the deformation of the spherical graphite precipitates and the formation of M_6_C and M_3_C_2_ carbide phases. These carbides are formed in a ferritic matrix or at the grain boundaries, resulting in increased hardness and a drastic reduction in impact strength. The article presents the influence of heat treatment on the material’s microstructure and resistance to abrasive wear. Chromium and aluminum additions can also indirectly reduce the abrasive wear resistance of SiMo cast iron. The presented research shows the possibility of doubling the abrasive wear resistance of SiMo cast iron.

## 1. Introduction

Medium-alloy ductile iron with a ferritic matrix of the SiMo type is comparable to white high-chromium cast iron in terms of oxidation resistance. The difference is that silicon–molybdenum ductile iron (SiMo) can be used to make complex shapes, e.g., the bodies of various types of turbines and turbine rotors, and exhaust gas collectors in diesel engines [1,2]. It should be noted that SiMo cast iron should have a ferritic matrix to eliminate the pearlite precipitates that occur at the grain boundaries; ferritising annealing is planned [3,4]. The purpose of ferritising annealing is to completely or partially disintegrate the M_3_C eutectoid carbide. Because this carbide is a thermodynamically unstable component of the structure, it decomposes into ferrite, austenite, and graphite during the long-term annealing of cast iron at high temperatures. The speed of the process depends on the temperature. The precipitation of pearlite at the grain boundaries is an unfavourable phenomenon; heat treatment may be necessary to eliminate the pearlite from the matrix. Perlite reduces the plasticity, which makes SiMo cast iron highly brittle when it is combined with the carbide mesh at the grain boundaries. The high silicon content in this type of cast iron partially neutralises the harmful effect of molybdenum on the structure and properties, increasing the ability to form ferrite in the cast iron. As a result, the plasticity increases. Due to its properties, the silicon means that the amount of carbon in cast iron in foundry practice does not exceed 3% by weight, especially with Si contents that are greater than 4.5% by mass [5,6,7,8].

From an economic point of view, it is advantageous to keep the heating time as short as possible while using the highest permissible value of the treatment temperature. For unalloyed grey cast iron, the temperature range is 760–820 °C (silicon and chromium can increase the Ac1 temperature), and the heating time should be approximately 60 min per 25 mm of wall thickness [4]. In [9], research was carried out to determine the heat-treatment temperature for SiMo iron castings. It is true that the carbides were dissolved at a temperature of 1100 °C; when the castings cooled, however, molybdenum was released again in the form of carbides with a different morphology at the grain boundaries. As a consequence, the plasticity of the material that was produced was much lower than that of the others. The tests showed that the best mechanical properties were obtained when heated at 950 °C and were slightly worse at 800 °C. However, a temperature of 800 °C is more acceptable to casting manufacturers from an economic point of view.

An analysis of the literature on abrasion wear indicated that the research often concerned Fe–C alloys in which the matrix was reinforced with carbides or nitrides or the castings had abrasion-resistant composite zones [10]. Another solution was to produce the primary carbides of vanadium, titanium, and niobium [11] in the entire volume of the molten steel during the metallurgical process, where, in abrasion-resistance tests, a much higher resistance to abrasive wear was achieved as compared to the reference samples.

In [12], it was shown that in the case of SiMo cast iron castings, the hardness and resistance to abrasive wear increase with increasing molybdenum content. The authors emphasise that this is most likely related to the formation of molybdenum carbides at the grain boundaries. In [13], the authors showed that the addition of chromium, vanadium, and nickel to SiMo cast iron improves resistance to abrasive wear. The authors see a beneficial effect of M_7_C_3_ alloy carbides and pearlite in SiMo cast iron with alloy additions. The paper presents the issue of the abrasive wear resistance of silicon–molybdenum ductile cast iron enriched with aluminium and chromium at ambient temperatures. This type of cast iron can be used in devices requiring high resistance to high-temperature corrosion. An example of such a casting is turbocharger housings operating in cycles from ambient temperatures to temperatures above 1000 °C. High-temperature corrosion products are deposited on the casting surface [14]. Regeneration of such a casting can be achieved by performing mechanical sandblasting, which is a special case of micro-cutting.

As an addition to cast iron, chromium increases its resistance to high temperatures [14,15,16,17,18]. As a component of nickel–chromium cast iron at an amount of up to 2%, it also has a positive effect on the hardness [19]. Chromium enhances the tendency of cast iron to metastable crystallisation with precipitation carbides, which may make it necessary to produce cast iron with an addition of aluminium. In high-chromium cast iron, the chromium forms M_7_C_3_ carbides, which results in this type of cast iron having good resistance to abrasive wear [20,21]. The addition of this element to SiMo cast iron can also cause chromium carbides to appear in the structure, improving its abrasion resistance [22].

Aluminium improves the graphitisation of cast iron [22,23,24]. With an aluminium content of 3.2–3.8%, the structure is ferritic [25]. The high silicon content raises the eutectoid transformation A1 temperature to approximately 900 °C. Molybdenum forms a carbide phase at the grain boundaries. The carbide mesh improves the dimensional stability and, above all, increases the tensile strength and creep resistance [26,27].

Molybdenum increases the temperatures of the phase transformations and increases their ranges [28,29]. In addition to the previously mentioned formation of carbides at the grain boundary in SiMo cast iron, other components can also segregate (such as magnesium [29,30]), creating phases that reduce the mechanical properties, that, when in combination with molybdenum carbides, will be unfavourable for castings that are made of this type of cast iron.

The purpose of the research was to determine the influence of the content of aluminum, chromium, and molybdenum in SiMo cast iron on the resistance to abrasive wear.

## 2. Research Methodology

The melts of ductile silicon–molybdenum cast iron that were used for the tests were carried out in the Experimental Foundry at the Faculty of Foundry of AGH University of Science and Technology in Krakow. The melts were made in a medium-frequency induction furnace with a crucible capacity of 15 kg. The chemical composition of the metal charge that was used during the melting is listed in Table 1. The molten metal was superheated to a temperature of 1550 °C and held at this temperature for 3 min. After this time, the process of spheroidising the cast iron was carried out using the bell method.

In this method, a steel container (bell) filled with a nodulant is introduced into the crucible with the molten metal on a special arm. A schematic diagram of the container (bell) that was used in this method is shown in Figure 1. For the nodularisation treatment, FeSiMg that contained 6.5% Mg was used in an amount that allowed for a magnesium content in the cast iron within a range of 0.03 ÷ 0.04%. Then, an inoculation was performed using an Elkem Zircinoc inoculant in an amount of 0.2% by mass. The composition of the inoculant is provided in Table 2. The nodularisation process lasted 5 min. The molten metal prepared this way was poured into moulds that were made of bentonite mass, reproducing the experimental castings of a Type-2 “Y” ingot with a wall thickness of 25 mm. The cast of sample Y is shown in Figure 2. 

The material that was obtained from the Y ingots was heat-treated to remove the pearlite from the matrix and attempt to dissolve the carbides that were located at the grain boundaries. For this purpose, the material was placed in a furnace at 820 °C and annealed for 3 h, followed by cooling in the furnace to an ambient temperature. A diagram of the heat-treatment process is shown in Figure 3.

Samples for testing microstructure, hardness, and abrasion resistance were cut from the casting shown in Figure 3. A chemical analysis of the cast iron samples from the test ingots was performed using a BRUKER TASMAN Q4 spectrometer that was manufactured at the Upper Silesian Institute of Technology of the Łukasiewicz Research Network. in Gliwice. The chemical composition of the tested cast iron is presented in Table 3.

Calculations of the crystallisation temperatures of the individual phases for the cast iron with a specific chemical composition were performed using the Thermo-Calc, program version 2019a (based on the CALPHAD method).

The microscope observations were made on a Leica MEF4M optical microscope, (Leica Microsystems, Wetzlar, Germany) using the Leica Q Win program (Version 6). 

Along with the chemical composition analysis, the SEM microstructure and phase composition analysis in the structure of the investigated cast iron were performed on a JEOL 500LV scanning microscope (JEOL Ltd., Tokyo, Japan) with an X-ray microanalysis (EDS) attachment.

Microanalysis investigations were performed with a probe-corrected Themis (200 kV) field emission gun (FEG) transmission electron microscope (TEM) that was equipped with a windowless four-quadrant Super-X detector of the integrated energy dispersive spectroscopy (EDS) system manufactured by the Thermo Fisher company, Waltham, MA, USA. The maps that presented the local chemical composition were built of 500 × 500 pixels (acquired with a 0.5 nm electron probe) using ‘net counts’; this helped to minimise the overlap of the neighbouring lines. Thin foils for these investigations were prepared with the focused ion beam (FIB) method using a Scios dual-beam (e−/Ga+) scanning micro, scope (also manufactured by Thermo Fisher). 

Samples for EDS testing were prepared so that the analysis surface covered the grain boundaries with carbides located there.

The Vickers hardness measurement was performed using an SBRV-100D hardness tester (Guizhou Sunpoc Tech Industry Co., Ltd., Guiyang City, China). The tribotester 3-POD stand that was developed at the Department of Foundry of the Silesian University of Technology was used to test the abrasive wear. This station was largely modelled on the solutions of stations that are classified as pin-on-disc wear-testing methods in which a tested rod-shaped sample (e.g., a square or circular cross section) is pressed with a given force P to a rotating disc that constitutes the counter sample, which causes the abrasive wear of the tested samples [31,32]. 

The mechanism of abrasive wear that occurs on the Tribotester 3-POD abrasive wear test stand used in this work can be described by three processes: microcutting, grooving, and scratching [15,33].

Such processes occur when, in the friction areas of interacting elements, there are loose or fixed abrasive particles or protruding irregularities of a harder material that act as microblades. The counter-sample in the research device used is sandpaper with hard-fixed silicon carbide (SiC) abrasive grains, causing intense abrasive wear mainly by the micro-cutting mechanism.

The device can test three samples simultaneously. The samples were divided into two sets: Set 1—Samples 0 and B; and Set 2—Samples A, C, and D. The samples were mounted in a special rotating holder that allowed for the individual pressure of each sample to be transferred to the abrasive disc by the use of linear guides. The abrasive disc was a sandpaper disc with a specific gradation (grain) and type of abrasive grain. A new abrasive disc was used for each set of samples, and the number of abrasion stages was determined within the established total abrasion path. After each stage, the samples were weighed.

The direction of the rotation of the abrasive disc was opposite to the direction of the rotation of the sample holder. The rotational speed of the abrasive disc (counter sample) was 92.0 rpm, and the rotational speed of the test sample holder was 87.5 rpm. The abrasion path was determined by the number of revolutions of the abrasive disc. Sample clamping force P resulted from the sum of the weight of the moving part of the test sample holder, the attached weight, and the weight of the test sample. Figure 4 shows a diagram (a) and a photo of the working part of the device (b)

The operating parameters of the 3-POD tribotester in the conducted tests were as follows: 

Counter sample (abrasive disc): sandpaper with SiC grains (80 grit); Test samples with nominal dimensions of 10 × 10 × 30 mm; Single sample load: 230, 300, and 380 G; Total abrasion distance on new sandpaper: 3000 m; Three stages of abrasion: Stage I—500 m; Stage II—1000 m, Stage III—1500 m (weighing after each stage); Dry abrasion.

## 3. Test Results and Analysis

### Modelling of Thermodynamic Parameters of Tested Cast Iron Using Thermo-Calc Program

Calculations were made using ThermoCalc to check which carbides and phases could be separated during crystallisation. 

Figure 5 shows the results of the thermodynamic calculations for the tested SiMo cast iron that contained 5.2% Si and 3% C.

It can be seen that, during the crystallisation of the cast iron, M_6_C carbide was released and stable at room temperature. In the case of the cast iron without the additions of Cr and Al (Figure 5), the first phase that crystallised from the molten metal at a temperature of 1210 °C was the graphite. This proved the hypereutectic nature of the alloy. The austenite appeared at a temperature of 1162 °C, while the M_6_C carbide phase appeared at 1160 °C (almost simultaneously with the austenite). The beginning of the eutectoid transformation was at a temperature of 980 °C; below a temperature of 920 °C, there was no more austenite in the cast iron. The simulation results for the cast iron with the same carbon and silicon contents yet with the addition of 1% Cr and 2% Al are shown in Figure 6.

In addition to the M_6_C carbide, the M_3_C_2_ carbide also crystallised in the case of this alloy (which was also stable at room temperature). After adding 1% Cr and 2% Al, the first phase to crystallise at 1280 °C was the graphite. The beginning of the austenite formation occurred at a temperature of 1212 °C; this indicated a significant increase in the liquidus temperature as compared to the previous sample that was discussed above. The M_6_C carbide crystallised at 1210 °C (i.e., its formation temperature was close to the formation temperature of the austenite). The M7C3 carbide was formed at a temperature of 690 °C, but it disappeared at 579 °C (the M_3_C_2_ carbide was formed in its place). The beginning temperature of the eutectoid transformation was 1050 °C, and its ending temperature was 960 °C. After increasing the amount of the Cr to 3% and the Al to 4% (Figure 7), no additional phase crystallisations could be observed in the cast iron (but the amounts of the M_6_C and M_3_C_2_ carbides increased).

The SiMo cast iron with the addition of 3% Cr and 4% Al was characterised by even higher crystallisation temperatures of the individual alloy phases. In this case, the graphite was also the first to crystallise (at a temperature of 1310 °C). The beginning of the eutectic crystallisation occurred at a temperature of 1230 °C, but the first M_6_C carbides crystallised earlier (at a temperature of 1240 °C). The M_7_C_3_ carbides appeared at 990 °C; below 640 °C, the M_3_C_2_ carbides appeared instead. The starting temperature of the eutectoid transformation was 1100 °C, and its ending temperature was 980 °C. Table 4 shows the chemical compositions of the carbide phases, as calculated by the Thermo-Calc programme.

## 4. Microstructures of Tested Cast Iron

### 4.1. Sample 0 (Reference)

The reference sample for testing the microstructure, hardness, and resistance to abrasive wear was made of EN-GJS-500-7-grade cast iron, according to the EN-1563:2018 standard [34]. The structure of the cast iron is shown in Figure 8. The structure was characterised by spheroidal graphite (approx. 90% of all of the graphite precipitates), which was evenly distributed throughout the volume of the cast iron, and a small amount of degenerate graphite. The matrix of this type of cast iron was ferritic and pearlitic, with the amount of ferrite being approximately 60% and the pearlite about 40%. It should be noted that GJS-500-7 cast iron does not contain elements such as Cr or Al in quantities that would lead to the degeneration of the spheroidal graphite precipitates.

### 4.2. Sample A

In the microstructure of the unetched specimen that is shown in Figure 9a, it can be seen that most of the graphite precipitates were in the form of spheres. The graphite was evenly distributed. About 20 to 25% of the graphite precipitates were degenerate. Some graphite precipitates in the form of interconnected balls indicate the low degree of mixing of the inoculant.

The microstructure of the etched specimen that is shown in Figure 9b shows a matrix that consisted of ferrite. Alloy carbides that contained molybdenum were visible at the grain boundaries; these were M_6_C carbides that were stable during the heat treatment. The distributions of the most important elements in the individual phases of the structure are shown in the EDS maps in Figure 10. It can be seen that the small amounts of chromium, manganese, silicon, and iron that were contained in Sample A were present in both the matrix and the carbide precipitates. The molybdenum was concentrated exclusively in the carbide precipitates.

### 4.3. Sample B

In the microstructure of Sample B, it can be seen that approximately 50% of the graphite precipitates were spheroidal graphite (Figure 11). The remaining 50% of the degenerated precipitates took the form of very fine graphite precipitates of various shapes. There were noticeable areas where the graphite was much larger than that in the rest of the structure; this was most likely the effect of the increased aluminium concentration, which led to the phenomenon of increased graphitisation in these areas of the molten metal. One can also notice the precipitation of the exploded graphite, which was related to the too-high magnesium content in the cast iron and the degenerative effect of the aluminium on the precipitation of the nodular graphite.

The microstructure of the etched specimen that is shown in Figure 11b shows a matrix that consisted of ferrite. A network of M_3_C_2_ alloy carbides was revealed at the grain boundaries, which were stable during the heat treatment. Moreover, small precipitates of alloyed carbides were formed in the ferrite grains. To determine the chemical compositions of the phases in the structure of the tested cast iron, an EDS analysis was performed (Figure 12). The results of the analysis for Area 5 and Point 6 are in Table 5 and Table 6. According to the analysis, the M_3_C_2_ carbides appeared in Area 5, while the MC carbides appeared at Point 6.

The surface distribution of the individual elements is shown in Figure 13. In the case of Sample B, chromium and molybdenum mainly formed carbides and were not dissolved in the matrix. In addition to iron, however, the matrix also consisted of silicon and aluminium.

### 4.4. Sample C

In the microstructure of the unetched specimen of Sample C (Figure 14), it can be stated that approximately 25% of the graphite precipitates were spheroidal graphite. Most of the graphite precipitates were vermicular; this was due to the too-low magnesium content, taking the graphite’s degenerate effect of the aluminium into account. The smaller amount of aluminium than in Sample B meant that there were no large graphite precipitations in the structure. The heat treatment in this case was intended to remove the carbides (which was not achieved). After the heat treatment, there were no small carbides inside the grains (which was the case in the structure of Sample B). It can be seen that the number of carbides was smaller than in the case of Sample B.

In the case of Sample C, an EDS analysis of the precipitates (marked in Figure 15) was also performed. The results of the analysis for Point 4 and Area 5 are presented in Table 7 and Table 8. In the case of Point 4, it was an MC-type carbide, and its morphology was similar to the MC carbide from Sample B. However, the analysis of Area 5 indicated that there were M_6_C carbides on the grain boundaries of Sample C.

The distribution of the elements in the structure is shown in Figure 16. In this case, it was clearly visible that the large carbide precipitates were complex carbides (but mainly molybdenum); on the other hand, the fine carbides were chromium carbides. In this case, aluminium and silicon were also dissolved in the matrix, and silicon was also present in the large carbides.

### 4.5. Sample D

In the microstructure of the unetched specimen of Sample D that is shown in Figure 17a, it can be seen that only about 10% of the graphite precipitates were spheroidal graphite; about 70% were vermicular graphite precipitates, and the remaining graphite precipitates degenerated in various ways. Carbide precipitates were visible at the ferrite grain boundaries, as was the case in Sample C (Figure 17b). The carbide precipitates in the discussed Sample D were not as extensive as those in Sample B.

The distribution of the elements in the structure of Sample D is shown in Figure 18. The precipitations at the grain boundaries were complex carbides. The fine carbide precipitates consisted of chromium and molybdenum. Aluminium was dissolved in the metal matrix, although a small spectrum of this element could also be seen in the carbides.

The characteristics of the microstructures of the samples are listed in Table 9. The table specifies the matrix, the types of the graphite precipitates, and the other phases that were present in the structure.

## 5. Results of Abrasive Wear Tests Using Pin-on-Disc Method

Table 10 contains the results of the abrasive wear tests of the tested samples that were carried out on the tribotester 3-POD stand. Additionally, Table 10 also shows the hardness of the samples (measured on the Vickers scale) and the total abrasive wear at a specific sample load. 

Figure 19, Figure 20, Figure 21, Figure 22 and Figure 23 show the abrasive wear graphs of all of the tested samples. Increases in abrasive wear could be observed as the abrasive distance was covered and the sample pressure on the abrasive disc increased. The abrasive wear rate decreased as the abrasive distance increased, which was justified by the wear of the abrasive disc (sandpaper). The graph in Figure 23 confirms the strong influence of the sample load on its wear. The calculations were made according to Formula (1):(1)$$R=\frac{∆m_{0}}{∆m_{B}}$$
where:

Δ*m*_0_—total abrasive wear of reference sample, g;

Δ*m_B_*—total abrasive wear of test sample, g.

Samples B, C, and D had relatively high hardness values (more than 350 HV) but only slightly higher abrasive wear resistance as compared to Reference Sample 0. Among these samples, only Sample B exceeded the relative abrasive wear resistance (R > 2).

The test result of Sample C is interesting, having a hardness greater than 350 HV showed worse resistance to abrasive wear when compared to Reference Sample 0 (whose hardness did not exceed 200 HV). Sample C had the lowest wear resistance (Figure 23) with the highest sample load during the abrasive wear measurements. The research showed that the tested SiMo medium alloy cast iron had a soft structure in the form of ferrite and numerous hard, complex carbides. During the abrasive wear, some of the carbides may have crumbled, especially under heavy loads. Therefore, this type of cast iron had a slightly lower relative resistance to abrasive wear. Furthermore, it can be observed that an appropriately selected heat treatment for Sample B doubled its relative resistance to abrasive wear when compared to the cast iron that made up Reference Sample 0.

## 6. Conclusions

The addition of aluminium and chromium to SiMo cast iron increased the amount of M_6_C, M_3_C_2_, and MC carbides; these remained stable during the proposed heat treatment and consequently increased the hardness of the cast iron.The addition of aluminium caused the degeneration of the spheroidal graphite precipitates.The highest abrasion resistance, more than twice the abrasion resistance of the reference sample with a ferrite–pearlite matrix, was obtained with the highest addition of Cr, Mo, and Al—sample B.Samples A and C with a ferritic matrix, having a smaller amount of carbides in the structure than sample B, had abrasion resistance comparable to that of the reference sample. Sample D had 19 to 38% better abrasion resistance than the reference sample.

## Figures and Tables

**Figure 1 materials-17-01223-f001:**
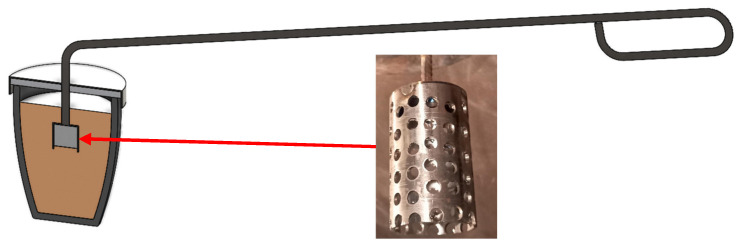
Steel container (bell) containing nodulant and inoculant used in the process of nodularisation of molten metal.

**Figure 2 materials-17-01223-f002:**
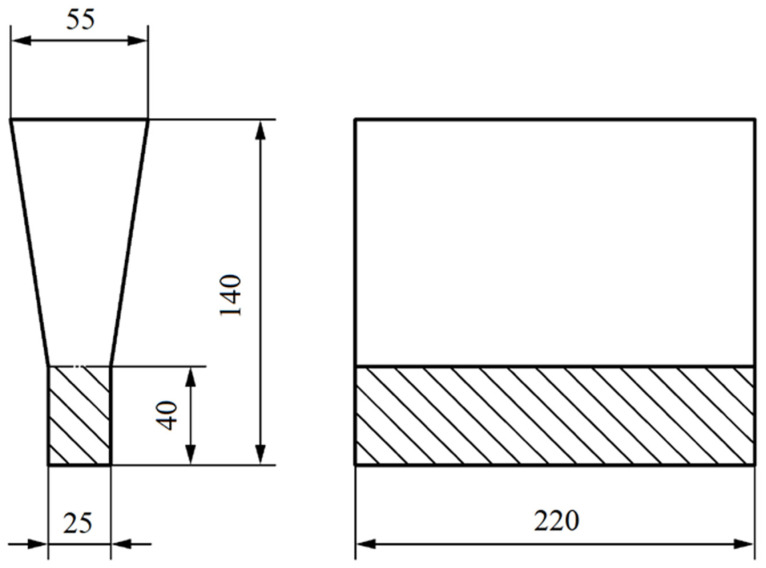
Type 2 Y ingot with marked test sampling areas and dimensions in mm.

**Figure 3 materials-17-01223-f003:**
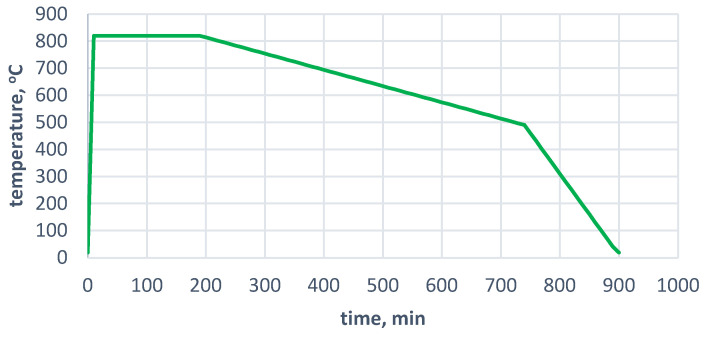
Scheme of ferritising annealing procedure that was used for SiMo cast iron carried out at METALPOL Węgierska Górka foundry.

**Figure 4 materials-17-01223-f004:**
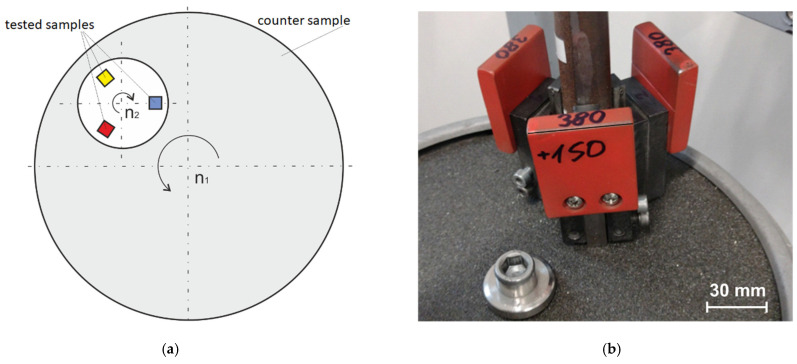
3-POD tribotester: (**a**) device operation diagram; (**b**) view of working system (research samples–counter sample).

**Figure 5 materials-17-01223-f005:**
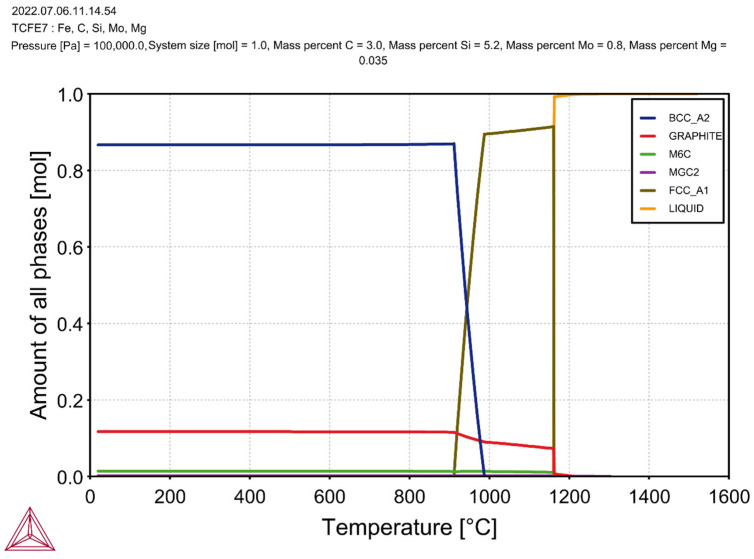
Simulation results for cast iron without addition of Cr and Al.

**Figure 6 materials-17-01223-f006:**
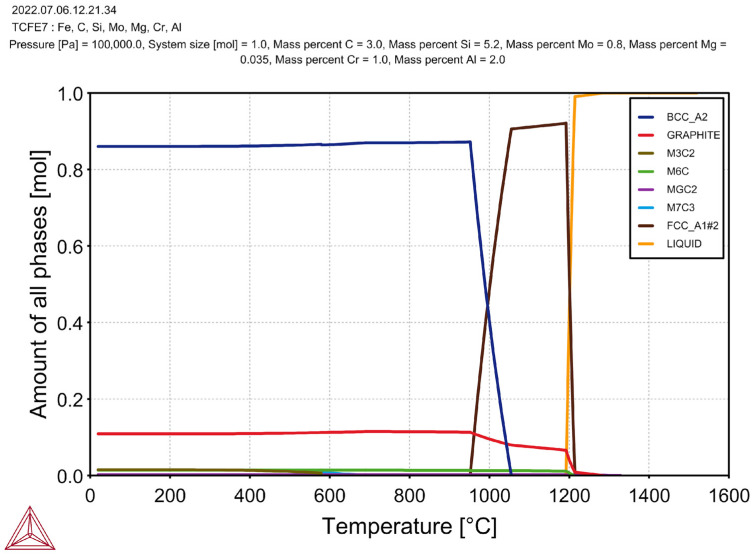
Simulation results for cast iron with addition of 1% Cr and 2% Al.

**Figure 7 materials-17-01223-f007:**
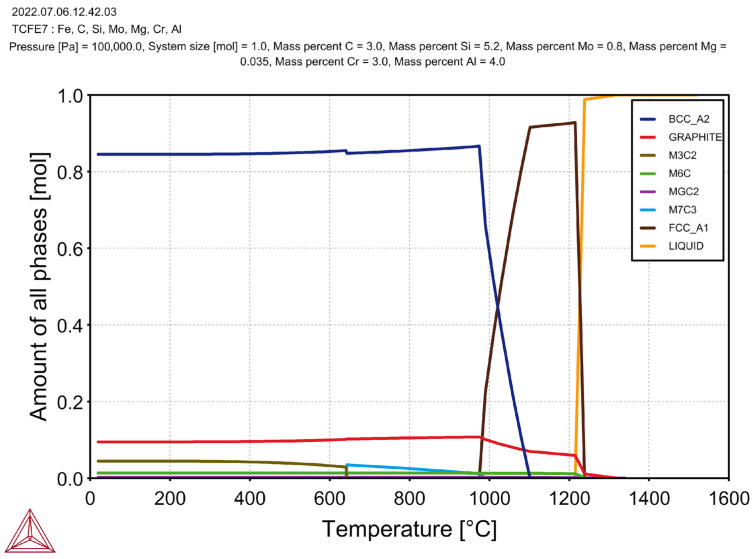
Simulation results for cast iron with addition of 3% Cr and 4% Al.

**Figure 8 materials-17-01223-f008:**
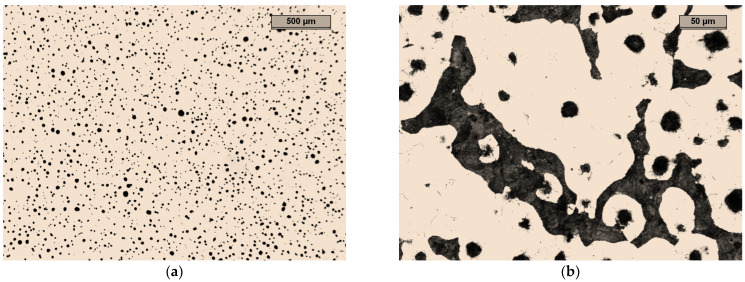
Microstructure of sample 0: (**a**) optical microscope: unetched, magnification 25×; (**b**) etched with nital, phases: light—ferrite, dark brown—perlite, black with a round shape—graphite, magnification 200×.

**Figure 9 materials-17-01223-f009:**
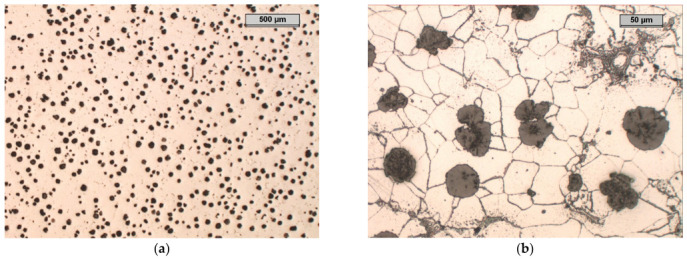
Microstructure of sample A, optical microscope: unetched, magnification 25× (**a**); etched with nital, phases: light—ferrite, shades of grey—carbides, black—graphite, magnification 200× (**b**).

**Figure 10 materials-17-01223-f010:**
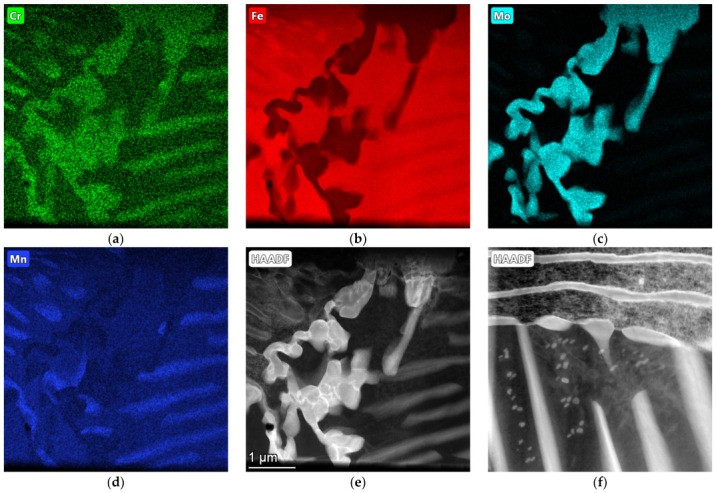
EDS maps of sample A, indicating concentrations of (**a**) chromium, (**b**) iron, (**c**) molybdenum, and (**d**) manganese; (**e**,**f**) HAADF imaging (transmission microscope).

**Figure 11 materials-17-01223-f011:**
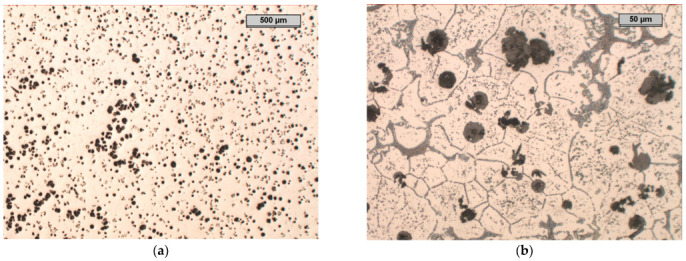
Microstructure of sample B, optical microscope: (**a**) unetched, magnification 25×; (**b**) etched with nital, phases: light—ferrite, shades of grey—carbides, black—graphite, magnification 200× (**b**).

**Figure 12 materials-17-01223-f012:**
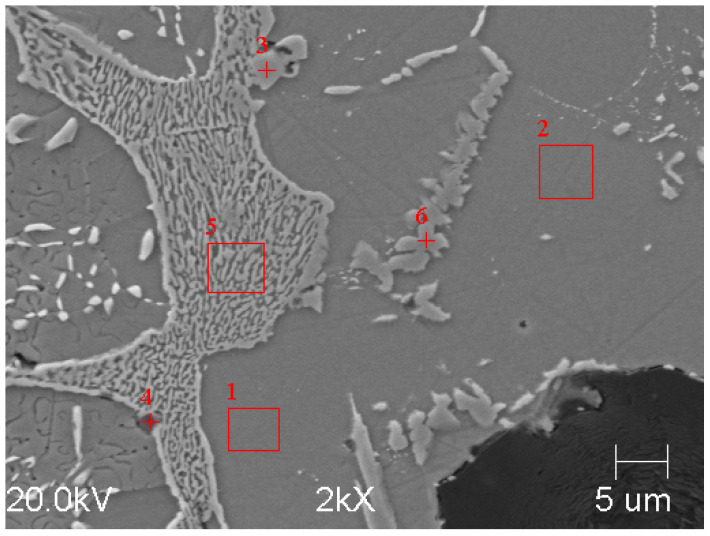
Microstructure of sample B, etched with nital, magnification 2000× (SEM), 1 and 2—metal matrix, 3, 4 and 6—carbides, 5—molybdenum carbide precipitates.

**Figure 13 materials-17-01223-f013:**
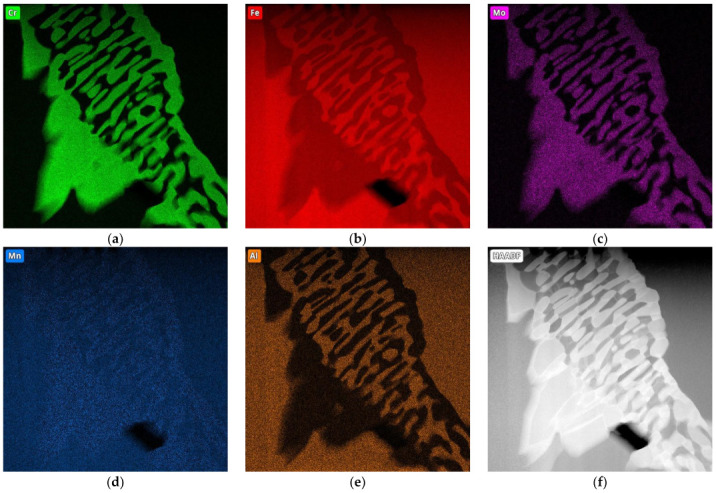
EDS maps of Sample B, indicating concentrations of (**a**) chromium, (**b**) iron, (**c**) molybdenum, (**d**) manganese, and (**e**) aluminium; (**f**) HAADF imaging (transmission microscope).

**Figure 14 materials-17-01223-f014:**
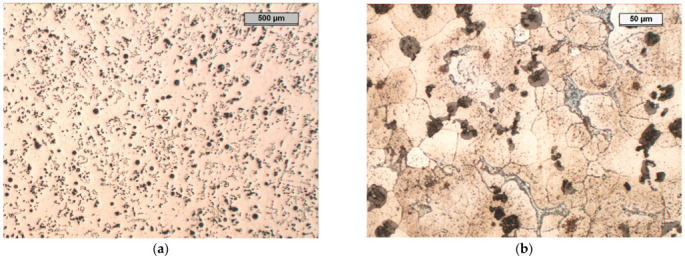
Microstructure of Sample C, optical microscope: (**a**) unetched, magnification 25×; (**b**) etched with nital, phases: light—ferrite, shades of grey—carbides, black—graphite, magnification 200× (**b**).

**Figure 15 materials-17-01223-f015:**
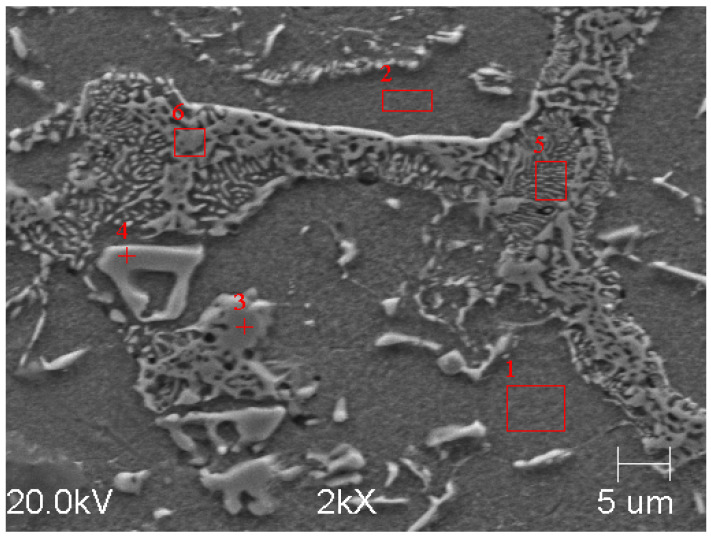
Sample C etched with nital, magnification 2000× (SEM), 1 and 2—metal matrix, 3 and 4—carbides, 5 and 6—molybdenum carbide precipitates.

**Figure 16 materials-17-01223-f016:**
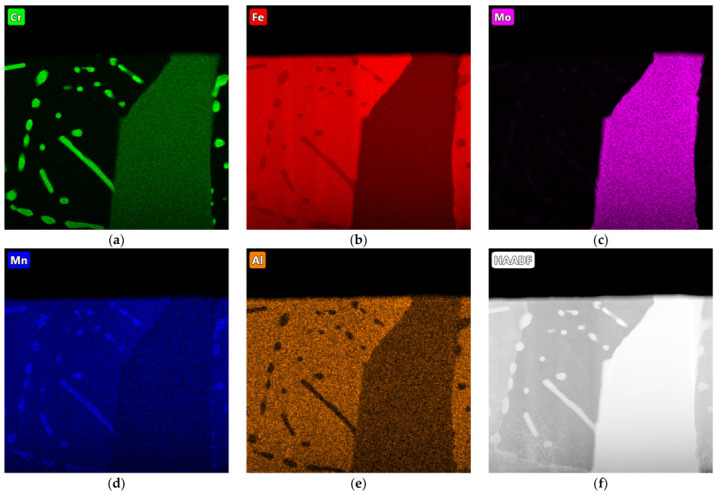
EDS maps of sample C, indicating concentrations of (**a**) chromium, (**b**) iron, (**c**) molybdenum, (**d**) manganese, and (**e**) aluminium; (**f**) HAADF imaging (transmission microscope).

**Figure 17 materials-17-01223-f017:**
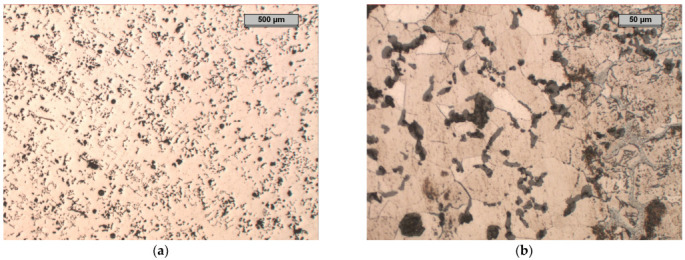
Microstructure of sample D, optical microscope: (**a**) unetched, magnification 25×; (**b**) etched with nital, phases: light—ferrite, shades of grey—carbides, black—graphite, magnification 200× (**b**).

**Figure 18 materials-17-01223-f018:**
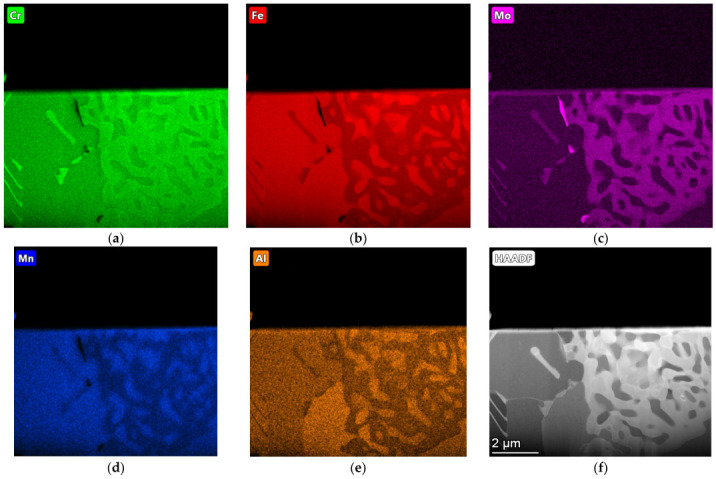
EDS maps of sample D, indicating concentrations of (**a**) chromium, (**b**) iron, (**c**) molybdenum, (**d**) manganese, and (**e**) aluminium; (**f**) HAADF imaging (transmission microscope).

**Figure 19 materials-17-01223-f019:**
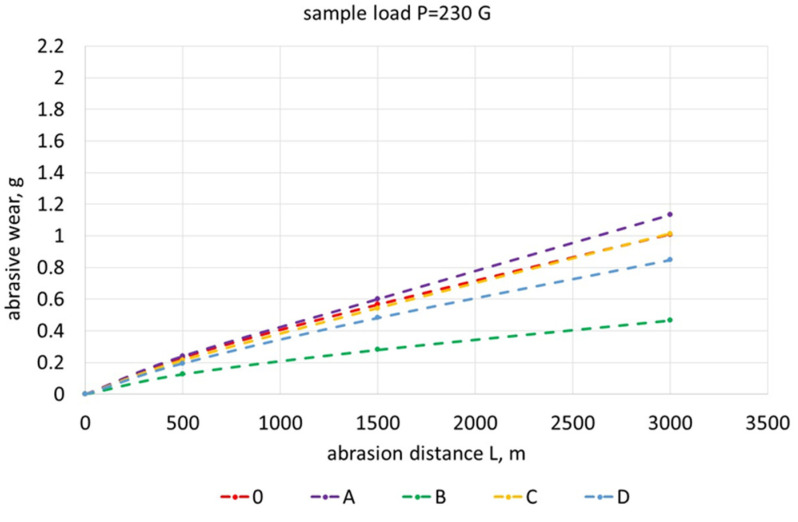
Abrasive wear of tested samples as function of abrasion path with sample load of 230 G.

**Figure 20 materials-17-01223-f020:**
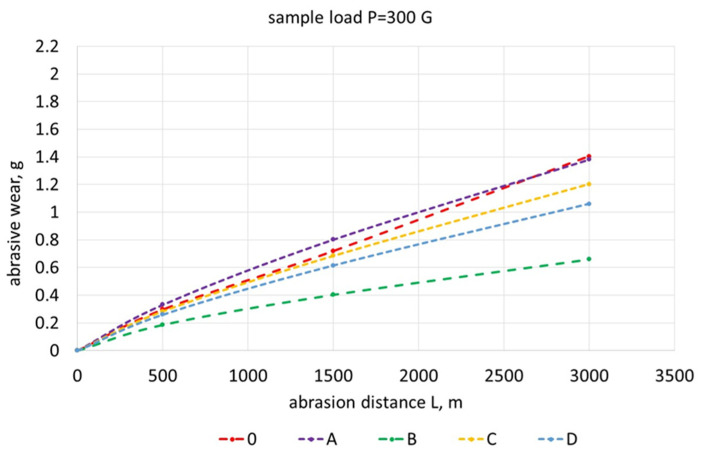
Abrasive wear of tested samples as function of abrasion path with sample load of 300 G.

**Figure 21 materials-17-01223-f021:**
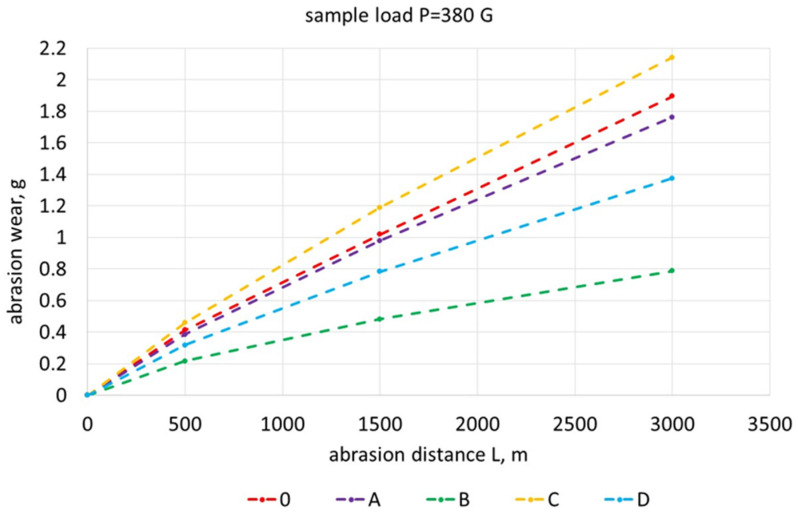
Abrasive wear of tested samples as function of abrasion path with sample load of 380 G.

**Figure 22 materials-17-01223-f022:**
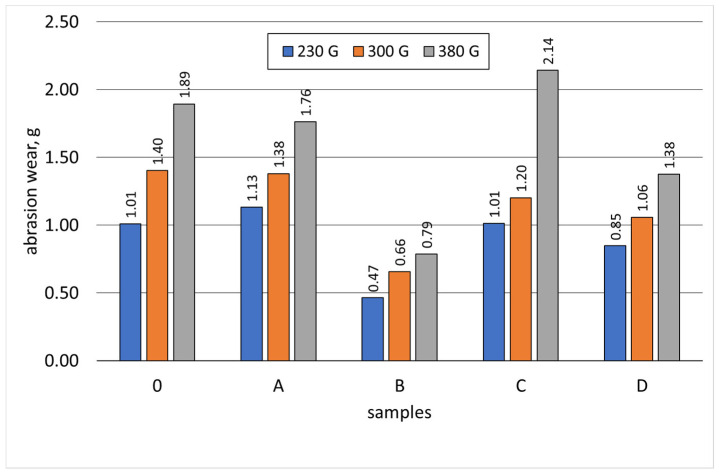
Effect of sample load on abrasive wear.

**Figure 23 materials-17-01223-f023:**
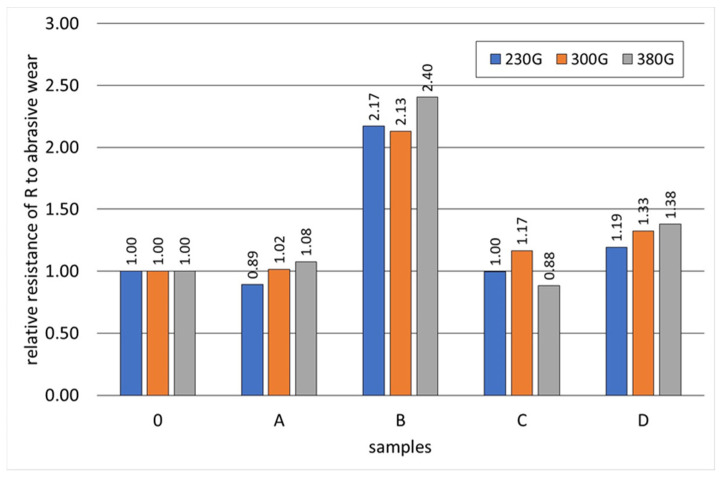
Relative resistance R to abrasive wear of tested samples (Reference Sample 0).

**Table 1 materials-17-01223-t001:** Chemical composition of charge materials used during cast iron melting.

Charge Material	C	Si	Mn	P	S	Mo	Cr	Al	Fe
	wt.%								
Foundry pig iron	4.44	0.81	0.02	0.025	0.009	-	-	-	rest
W3 silicon steel scrap	0.02	1.60	0.37	0.074	0.01	-	-	-	rest
W3 manganese steel scrap	0.25	0.25	0.40	0.009	0.02	-	-	-	rest
Cast iron scrap	3.50	2.60	0.35	0.035	0.004	-	-	-	rest
FeSi	0.12	75.21	-	0.017	0.004	-	-	-	rest
FeMo	0.20	0.30	-	-	-	68.0	-	-	rest
FeCr	-	-	-	-	-	-	65.0	-	rest
Al	-	-	-	-	-	-	-	100.0	rest
Carburizer 9904	99.59	-	-	-	0.020	-	-	-	

**Table 2 materials-17-01223-t002:** Chemical composition of Zircinoc inoculant.

Inoculant	Si	Zr	Ca	Al
	%mas.			
Zircinoc	73.0–78.0	1.3–1.8	2.0–2.5	1.0–1.5

**Table 3 materials-17-01223-t003:** Chemical composition of tested cast iron.

Melts	Chemical Composition, % mas.											
	C	Cr	Cu	Mg	Mn	P	S	Si	Ti	Mo	Al	Fe
0	3.56	0.016	0.176	0.037	0.416	0.033	0.009	2.741	0.011	-	0.006	rest
A	2.67	0.099	0.095	0.031	0.384	0.021	0.008	5.56	0.019	0.81	0.023	rest
B	2.66	2.18	0.115	0.047	0.475	0.045	0.012	5.91	0.024	1.19	3.41	rest
C	2.60	0.97	0.126	0.034	0.455	0.047	0.013	5.61	0.020	1.02	1.66	rest
D	3.13	0.092	0.108	0.038	0.461	0.032	0.007	5.27	0.020	0.78	1.48	rest

**Table 4 materials-17-01223-t004:** Chemical compositions of carbide phases calculated by Thermo-Calc program.

Phase		Chemical Composition, % mas.				
	C	Si	Mo	Mg	Cr	Fe
M_6_C	3.2	15.1	51.6	-	-	rest
M_3_C_2_	13.3	-	-	-	86.6	rest

**Table 5 materials-17-01223-t005:** Quantitative EDS analysis of Area 5 of Sample B.

Chemical Composition										
% mas.										
C	Mg	Al	Si	P	S	Ti	Cr	Mn	Fe	Mo
10.967	0.122	0.890	4.339	0.000	0.000	0.144	9.376	0.598	66.036	7.529

**Table 6 materials-17-01223-t006:** Quantitative EDS analysis of Point 6 of Sample B.

Chemical Composition										
% mas.										
C	Mg	Al	Si	P	S	Ti	Cr	Mn	Fe	Mo
19.791	0.229	1.612	4.917	0.070	0.888	0.000	5.219	0.144	66.431	0.699

**Table 7 materials-17-01223-t007:** Quantitative EDS analysis from Point 4 of Sample C.

Chemical Composition										
% mas.										
C	Mg	Al	Si	P	S	Ti	Cr	Mn	Fe	Mo
15.358	0.091	0.706	4.626	0.065	0.000	0.291	2.788	0.146	74.697	1.233

**Table 8 materials-17-01223-t008:** Quantitative EDS analysis from Area 5 of Sample C.

Chemical Composition										
% mas.										
C	Mg	Al	Si	P	S	Ti	Cr	Mn	Fe	Mo
6.221	0.138	1.739	5.828	0.000	0.441	0.047	0.693	0.250	83.484	1.160

**Table 9 materials-17-01223-t009:** Characteristics of structures of tested samples.

Sample	0	A	B	C	D
Metal matrix	Ferritic-pearlitic	Ferritic	Ferritic	Ferritic	Ferritic
Graphite precipitation	Nodular graphite (>90%)	Nodular graphite (>75%)	Nodular graphite (approx. 50%)	Nodular graphite (approx. 25%) + compacted graphite (approx. 75%)	Nodular graphite (approx. 10%) + compacted graphite (approx. 70%)
Carbides	-	M_6_C carbides at grain boundaries	M_3_C_2_ carbides and MC at grain boundaries	M_6_C carbides and MC at grain boundaries	M_6_C carbides and MC at grain boundaries

**Table 10 materials-17-01223-t010:** Abrasive wear test results (total abrasive wear and resistance R) and Vickers hardness.

Set	Sample	Hardness HV	Sample Load, G			Sample Load, G		
			230	300	380	230	300	380
			Total Abrasive Wear, g			Relative Resistance, R		
1	0	197	1.010	1.403	1.895	1.00	1.00	1.00
	B	412	0.465	0.658	0.788	2.17	2.13	2.40
2	C	355	1.013	1.203	2.143	1.00	1.17	0.88
	D	362	0.848	1.058	1.375	1.19	1.33	1.38
	A	250	1.132	1.381	1.763	0.89	1.02	1.08

## Data Availability

Data are contained within the article.

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
