# Peer review of "Evaluation of Microstructure and Abrasive Wear-Resistance of Medium Alloy SiMo Ductile Cast Iron"

_materials, 2024, doi:10.3390/ma17051223_

Round 1

Reviewer 1 Report

Comments and Suggestions for Authors

The authors presented the impact of Cr and Al on the structure of SiMo cast iron especially their impact on the deformation of the spherical graphite precipitates and the formation of M6C and M3C2 carbide phases. I would recommend it to publish after minor revision.

(1) The introduction needs to be reorganized to reflect structural changes and to clearly review the motivation for this work.

(2) Experimental pictures need to be replaced with actual photos rather than schematic diagrams.

(3) The ruler in the figure needs to be enlarged, such as Fig. 8

(4) What is the black color in microstructure figures? Impurities or phases? such as Figs. 8-10.

(5) There is still a lack of analysis from aspects of wear mechanism and control conditions? Additionally, this article lacks insights and comments on the wear behavior, which should be included.

Comments on the Quality of English Language

Minor editing of English language required

Author Response

I am sending the responses to the review in the attachment.

Reviewer 2 Report

Comments and Suggestions for Authors

This work provides a thorough and insightful analysis of the impact of chromium and aluminum additions on the microstructure and properties of SiMo ductile cast iron. The discussion on the trade-off between increased hardness and decreased impact strength is particularly well-handled. However, the exploration of how chromium and aluminum indirectly affect wear resistance seems somewhat underdeveloped. The author is encouraged to delve deeper into this aspect in future research. Overall, the article offers valuable insights into the study of SiMo ductile cast iron, especially in terms of heat resistance and wear resistance.

The paper is recommended for acceptance, pending minor revisions to address any clarification or additional information that may be needed in certain sections.

1.      There are some mistakes like Cr and Al, Chromium and aluminum, in Abstract. The first time an abbreviation appears.

The serial numbering of the figures is inaccurate, and the first occurrence is actually Figure 2, which is too confusing; please check the entire manuscript.

2.  The analysis of the wear mechanism of different samples is not convincing, and to be precise, the authors did not provide a reasonable analysis of the wear mechanism, which can be referred to this literature to analyze the wear mechanism of the samples. 10.1002/adem.202001351, 10.1016/j.wear.2023.204864

3. Introduction- The authors fail to cite many studies on cast iron.

Please see the work of 10.1016/j.actamat.2012.04.042. There are many citations that could be included.

Author Response

(The authors gave the same response as above.)

Reviewer 3 Report

Comments and Suggestions for Authors

Paper: Evaluation of microstructure and abrasive wear-resistance of medium alloy SiMo ductile cast iron present few interesting experimental results in the metallurgical field of cast-irons. The paper is more suitable for Metals journal. The paper presents some major problems related to paper presentation, figures whitout technical novelties or figures unexplained in text.  

The figures starts with Figure. 2 Diagram of a steel container (bell) used in spheroidisation method, why is figure 1 after 3 ?, also Figure 2 can be eliminated 

A reference for Table 2 is necessary 

mention the technical value information of Figure 3 or eliminate the figure 

L142: number of chemical composition determinations and detector/equipment error plus each element standard deviation results , how reliable is the third digit ? 

model of EDS detector and the analysis mode: Automatic or Element List 

a scale in figure 4 b) should be inserted 

explain in text  HAADF imaging results presented in the entire paper  

why is Si mapping is missing from figures : 10, 14 and 18+20 - Si presence is confirmed in the energy spectrum 

how the authors determine: The characteristics of the microstructures of the samples are listed in Table 9.  The table specifies the matrix, the types of the graphite precipitates, and the other  phases that were present in the structure ? using what equipment or reasoning, please present references if necessary 

L506-510 : re-phrase 

Re-phrase first 3 conclusions paragraphs 

Comments on the Quality of English Language

Moderate editing of English language required

Author Response

(The authors gave the same response as above.)

Reviewer 4 Report

Comments and Suggestions for Authors

The paper "Evaluation of microstructure and abrasive wear-resistance of medium alloy SiMo ductile cast iron" presents valuable research with practical implications for materials with more wear resistance. I have following comments:

- The introduction should end with problem statement and objectives

- Figures throughout the article can be significantly improved. Please label all figures to mention what is what. The captions should be detailed and might repeat some information from article body to stand alone.

- Fig 3, all dimentions in mm?

- On page 5, why is it Fig. 1 again? also the text style in it doesn't match the rest of the article.

- In section 3, you have given calculation of some alloys while have left others, why? Maybe you can include all of them in Appendix. Also it would be beneficial if you use the alphabets A,B,C or so on for a certain alloy throughout the article.

- On top of Fig. 5, 6, and 7, I see that you have included major allying elements in ThermoCalc calculations while have intentionally left several micro alloying elements. How much does that affect the results?

- In explanation of Fig. 8 , you mention that there is pearlite in this material, where? I do not see any pearlite. Also in other results, I do see alot of pearlite but you did not mention it in the article body. The table 9 summarizes the results you have presented but it is opposite to what I see. Ferrite and graphite in 0, Ferrite and some carbide in A, but then Ferrite and pearlite in B, C, D.

- it would be great of you can label all figures in Section 4, and somehow construct them into one nice schematic to avoid repetition of similar images across the article.

- Providing EDS images is a nice idea, but you have to mention where where these measurements made on the sample. Are we lookin inside a grain or on the grain boundary.

- Similarly label SEM images with what we are looking at.

- Data on Fig. 13 is not readable at all.

- I see that the sample B has the highest hardness and then corresponding highest wear resistance. I do not understand the conclusion 4, the hardness does not decrease, looking at microstructural images, the pearlite is not eliminated, hardness of sample is not highest.

- Most of the references are more than 5 years old. Please avoid citing work that is older than 2019 unless it is immensely important.

- The conclusions need to be improved.

Comments on the Quality of English Language

The language can be improved, there are some grammatical and punctuation errors. 

Author Response

(The authors gave the same response as above.)

Round 2

Reviewer 3 Report

Comments and Suggestions for Authors

L108-110: re-phrase : additives 

L31: Introduction is 1 than at line L11: should be 2 not 1 

L131: in table 1 were is Fe ? should be mentioned , if exist, as rest or balance 

L135: a scale in the detail image is necessary 

L130: please clarify: The cast of sample Y is shown in Figure 3. or in figure 3 is: Scheme of ferritising annealing procedure .... 

In figure 2 the dimensions are in ? mm, cm ? 

L150: Re-phrase: Samples were made from the casting that is shown in Figure 3 to test the microstructure, hardness, and abrasion resistance. 

L157: Why is not mentioned Fe in Table 3 ?? is not mentioned in text either 

L177: how were followed the grain boundaries in the cutting process ? maybe a re-phrase is necessary: The test samples were cut at the grain boundaries where carbide precipitates occurred.  

Reviewer 4 Report

Comments and Suggestions for Authors

I agree with most of the changes, please delete Fig 13 and 17 by stating the observations in the article body.

The conclusions are still not adequate.
